# Use of CT, ED presentation and hospitalisations 12 months before and after a diagnosis of cancer in Western Australia: a population-based retrospective cohort study

Ninh Thi Ha [ORCID],[1] Sviatlana Kamarova [ORCID],[1,2,3] David Youens [ORCID],[1,4] Chau Ho [ORCID],[1] Max K Bulsara [ORCID],[4,5] Jenny Doust,[6] Donald Mcrobbie,[7] Peter O'Leary,[1,8,9] Cameron Wright,[1,10,11,12] Richard Trevithick,[13] Rachael Moorin [ORCID][1,4]

For numbered affiliations see end of article.

**Correspondence to**
Dr Ninh Thi Ha;
ninh.ha05@gmail.com

## ABSTRACT

**Objective** To examine the use of CT, emergency department (ED)-presentation and hospitalisation and in 12 months before and after a diagnosis of cancer.

**Design** Population-based retrospective cohort study.

**Setting** West Australian linked administrative records at individual level.

**Participants** 104 009 adults newly diagnosed with cancer in 2004–2014.

**Main outcome measures** CT use, ED presentations, hospitalisations.

**Results** As compared with the rates in the 12th month before diagnosis, the rate of CT scans started to increase from 2 months before diagnosis with an increase in both ED presentations and hospitalisation from 1 month before the diagnosis. These rates peaked in the month of diagnosis for CT scans (477 (95% CI 471 to 482) per 1000 patients), and for hospitalisations (910 (95% CI 902 to 919) per 1000 patients), and the month prior to diagnosis for ED (181 (95% CI 178 to 184) per 1000 patients) then rapidly reduced after diagnosis but remained high for the next 12 months. While the patterns of the health services used were similar between 2004 and 2014, the rate of the health services used during after diagnosis was higher in 2014 versus 2004 except for CT use in patients with lymphohaematopoietic cancer with a significant reduction.

**Conclusion** Our results showed an increase in demand for health services from 2 months before diagnosis of cancer. Increasing use of health services during and post cancer diagnosis may warrant further investigation to identify factors driving this change.

## STRENGTHS AND LIMITATIONS OF THIS STUDY

⇒ The present study examines (1) the patterns of hospital and emergency department services in conjunction with (2) both pre and post diagnostic periods of CT use, which has not been evaluated to date in adult patients with cancer.

⇒ The cohort makes use of whole of population data which removes selection bias and minimises loss to follow-up, while allowing for grounded and representative conclusions for the target population based on higher statistical precision.

⇒ The present data does not include some important clinical data such as indications for CT scanning and stage of cancer at diagnosis. Some CT scans from secondary hospitals may be missing.

## BACKGROUND

CT is useful for screening for cancer; diagnosis of the presence of a tumour and staging of cancer; guiding where to perform a biopsy procedure, where to apply local treatment (eg, radiofrequency ablation, implantation of radioactive seeds or cryotherapy); planning surgery, directing treatment strategies and assessing treatment response.[1–4] Examples of the utility of CT include the National CT Colonography Trial, where CT colonography provided similar accuracy in the rate of identifying both large colorectal polyps and colorectal tumours in patients with an average risk of colorectal cancer as compared with the gold standard colonoscopy.[1] Similarly in the National Lung Screening Trial, compared with chest X-ray, screening by CT reduced the risk of mortality from lung cancer in heavy smokers by 20%.[2] However, compared with standard X-ray (and non-ionising radiation modalities such as ultrasound and MRI), CT scanning imparts a relatively high radiation exposure that has raised concerns about the risk of cancer, particularly in children, young adults and women.[5–8] In addition to the radiation burden, increased use of CT scanning has also been associated with increases in incidental findings and overdiagnosis.[9 10] There

is therefore a trade-off between the benefits and risk of CT scanning which has led to calls to evaluate its use to ensure maintenance of high value care.

CT scanning has been steadily increasing in most developed countries, particularly in cancer care.[11–16] Studies in the UK[17] and Denmark[18] reported a significant increase in CT scanning from 3 months before a cancer diagnosis. A study by Purdie *et al* in New South Wales, Australia observed a similar pattern.[19] However, this study included only a small number of participants (n=894) and was limited to the period before a cancer diagnosis. While patterns of visits/consultation by general practitioners (GPs) or specialists before and after a cancer diagnosis have been examined previously,[20 21] patterns of emergency department (ED) presentations and hospitalisations in Australia remained unclear. Thus, in the present study, we examined changes in the use of CT, ED presentations and hospitalisations in the 12 months before and after a diagnosis of cancer for adults diagnosed with cancer between 2004 and 2014 in Western Australia (WA) using linked administrative data.

## METHODS
### Study design
This population-based retrospective cohort study follows the REporting of studies Conducted using Observational Routinely-collected health Data statement.[22]

### Data sources and linkage
This study is part of a larger project using person-level linked administrative data pertaining to a cohort comprising all Western Australians aged 18+ who had a separation (discharge) from any WA hospital for any non-pregnancy-related condition, or who presented for any reason to any public hospital ED, or who had a CT scan undertaken in WA by a public or private provider between 1 January 2003 and 31 December 2015.

Using state-based data from the WA Data Linkage System[23] and Medicare Benefits Schedule (MBS) data provided by the AIHW, cross-jurisdictional (ie, State – Commonwealth) privacy-preserving record linkage was undertaken by the Curtin University Centre for Data Linkage.[24] The following person-level linked data were available:

1. WA Cancer Registry data containing all incident cancers in the cohort (irrespective of age at diagnosis) from 1972 onwards, including diagnosis date, basis of diagnosis, site code, morphology code, behaviour type, cancer type code using the International Classification of Diseases, 10th revision, Australian Modification[25] and demographics (eg, age at diagnosis, sex and postcode).
2. WA Hospital Morbidity Data Collection records from 2003 to 2015 for all discharges from any WA hospital (public or private) for all conditions excluding pregnancy. It included dates and mode of admission and discharge, hospital type (tertiary or secondary), diagnosis (principal and co-diagnoses) and procedure codes.
3. WA Emergency Department Data Collection (EDDC) records 2003–2015 for all WA public hospital ED presentations. It included presentation date and time, presentation type (emergency or planned), triage code, diagnosis code and major diagnostic group.
4. WA Picture Archiving and Communication System (PACS) records for all CT scans (excluding positron emission tomography (PET)/CT) 2003–2015 undertaken in all WA public tertiary hospitals and selected public secondary hospitals (limited to hospitals who do not contract out radiology services) as an outpatient, ED patient or admitted patient. It included the scan date, the type of CT examination (eg, non-contrast head CT), and referral source.
5. MBS data capturing all CT scans subsidised by the Federal Australian Government (ie, those performed outside of hospital or in hospital for private patients), including the date and scan type, between 2003 and 2015.
6. WA Death Registrations records from 2003 to 2015 including the age and date of death.

### Study population
The study population was limited to patients aged 18+ years who were diagnosed with a first invasive cancer between 1 January 2004 and 31 December 2014. We excluded individuals who had a diagnosis of cancer after death or those who died within 30 days of a diagnosis of cancer in the present analysis.

### Patient and public involvement
Patients were not directly involved in the study design or conduct. Our consumer representative (Mr John Stubbs) was involved in the design of the grant used to fund this research and is an ongoing member of the research team providing ongoing input to analysis of the data, interpretation of the results and development of publications. The Western Australian Data Linkage Branch and the data custodians of the WA EDDC and the PACS data provided data for this project.

### Outcome measurement
The primary outcome was a number of CT scans performed within 12 months prior and 12 months post cancer diagnosis. Multiple CT scans within the same day of the same anatomical area were counted as a single CT event.[26] Similarly, we measured the number of ED presentations and hospitalisations within 12 months prior to and post cancer diagnosis. Numbers of CT scans, hospitalisations and ED presentations were captured monthly and annually for all cancer diagnosis in the study period from 2004 to 2014.

### Covariates
Cancer types were grouped as either solid or all lympho-haematopoietic (ALH) to examine the effect of this broad classification on outcomes of interest. Demographic

characteristics included sex, age group at cancer diagnosis and Indigenous status (captured using Data Linkage WA's Derived Indigenous Flag).[27] Accessibility to services was measured using the Accessibility/Remoteness Index of Australia[28] based on postcode at the diagnosis date. Socioeconomic status was classified using the Socio-Economic Indexes For Areas – Index of Relative Socio-economic Disadvantage based on postcode, for the census closest to diagnosis date categorised as quintiles.[29] Secondary hospitals in the suburbs of Bentley and Armadale did not contribute data on CT scans to PACS, as they contract out their radiology services to private providers, although data for CT scans from out-of-hospital and those attending tertiary hospitals who live in these suburbs were provided. Hospital catchment area was stratified as 'Bentley and Armadale' and 'others' to adjust for any potential bias in the regression model. Comorbidity was determined using the Multipurpose Australian Comorbidity Scoring System (MACSS)[30] capturing the number of 17 MACSS conditions recorded on any hospitalisation across the 5 years prior to cancer diagnosis. Referral sources of CT scans were grouped as ED physician, inpatient and out-of-hospital. Cancer diagnosis was also classified as either pathology or non-pathology-based diagnosis.

### Statistical analysis

Descriptive analyses were used to summarise the study population characteristics at the beginning of the study (2004), the end of the study (2014) and across the whole study period (2004–2014). After assessing the data for suitability (overdispersion) and model fit (based on Akaike and Bayesian information criterion) multivariable negative binomial regression was performed to calculate the monthly rates of CT scanning, ED presentations and hospitalisation 12 months prior to and 12 months post cancer diagnosis adjusted for all observed socio-demographic and clinical characteristics and year of diagnosis. Similarly, monthly rates of CT scanning, ED presentations and hospitalisation 12 months prior to and post cancer diagnosis were conducted for subcancer types. Multivariable negative binomial regression was also used to examine trends in total CT scanning use, ED presentations and hospitalisations in the year prior to and after diagnosis over the study period (2004–2014) adjusted for all observed characteristics and year of diagnosis. Analyses used Stata SE V.15.[31]

### RESULTS

### Characteristics of the study population

A total of 104 009 individuals had a first diagnosis of cancer from 2004 to 2014 (table 1). In table 1, the majority of the population were men (56.2%), aged 46–65 years (40.5%), classified as living in an area of least socioeconomic disadvantage (31.2%), lived in major cities (68.8%) and had at least one comorbidity (95.3%). In terms of cancer diagnosis, the most common type was solid cancer (91.2%) in which cancer of the male genital organs, digestive organs, melanoma and breast shared high proportions of 18.6%, 17.5%, 15.7% and 13.7% correspondingly. Up to 97% of cancer diagnoses were based on pathological methods. Both demographic and clinical characteristics were similar between the beginning (2004) and end of the study (2014).

### Use of CT scanning 12 months before and after cancer diagnosis and trends over the study period

The adjusted rate of CT scanning 12 months before and after cancer diagnosis is presented in table 2. At 12 months prior to cancer diagnosis, the adjusted rate of CT scanning was 14 CT scans per 1000 patients (95% CI 13 to 15) and increased to 35 CT scans per 1000 patients (95% CI 33 to 36) at 3 months before diagnosis. The rate steeply increased to 294 CT scans per 1000 patients (95% CI 290 to 298) in the month before diagnosis and reached the highest rate of 477 CT scans per 1000 patients (95% CI 471 to 482) in the month of diagnosis. After cancer diagnosis, the rate of CT scanning dramatically reduced but remained elevated compared with the rate pre diagnosis (further details of regression model output is presented in online supplemental appendix A). Similar patterns were observed across subcancer types as shown in online supplemental appendix B.

Figure 1 demonstrates the pattern of CT scanning 12 months before and after cancer diagnosis by two main cancer groups for two time points 2004 and 2014 for simplicity. While overall the rate of CT scanning is higher in patients diagnosed with ALH cancer than solid cancer, the patterns of CT use observed across the period 12 months pre and post cancer diagnosis were similar for both groups. For both years a peak in CT use occurred in the month of diagnosis and the peak was more pronounced for ALH cancers than solid cancer.

Further details in online supplemental appendix A shows that patients with a higher number of comorbidities were more likely to have CT scanning with an Rate Ratios (RR) 4.94 (95% CI 4.62 to 5.27) for those with at least six comorbidities versus those without any comorbidities. The rate of CT scanning stratified by gender, age, socioeconomic, accessibility to services, hospital catchment regions, cancer diagnosis methods was slightly different as compared with the reference group in each category.

Figure 2 panel A presents trends in total CT scans used in 12 months before and after diagnosis by cancer groups over the study period from 2004 to 2014. For 12 months before diagnosis, the rate of CT used in ALH cancer was consistently higher than those of solid cancers diagnosed over the entire study period. However, the rate of scans 12 months after diagnosis of ALH cancer significantly reduced between 2004 and 2009, following which the difference in the magnitude of the rate of CT scanning across the two cancer types disappeared (further details of regression model output is presented in online supplemental appendix C and D).

**Table 1** Characteristics of the study population

| Characteristics | | 2004 n=8081 | | 2014 n=10884 | | 2004–2014 (overall) n=104009 | |
|---|---|---|---|---|---|---|---|
| | | n | % | n | % | n | % |
| Sex | Male | 4468 | 55.29 | 5977 | 54.92 | 58411 | 56.16 |
| | Female | 3613 | 44.71 | 4907 | 45.08 | 45598 | 43.84 |
| Age group at diagnosis | 18–45 years | 909 | 11.2 | 1106 | 10.16 | 10827 | 10.41 |
| | 46–65 years | 3243 | 40.13 | 4283 | 39.35 | 42104 | 40.48 |
| | 66–75 years | 2008 | 24.85 | 2980 | 27.38 | 26142 | 25.13 |
| | >75 years | 1921 | 23.77 | 2515 | 23.11 | 24936 | 23.97 |
| Socioeconomic status* | Least disadvantage | 2419 | 29.93 | 2939 | 27.00 | 32464 | 31.21 |
| | Less disadvantage | 1224 | 15.15 | 1673 | 15.37 | 15295 | 14.71 |
| | Moderate disadvantage | 1939 | 23.99 | 1972 | 18.12 | 21954 | 21.11 |
| | High disadvantage | 1664 | 20.59 | 2941 | 27.02 | 23913 | 22.99 |
| | Highest disadvantage | 764 | 9.45 | 1351 | 12.41 | 9893 | 9.51 |
| | Unknown | 71 | 0.88 | 8 | 0.07 | 490 | 0.47 |
| Accessibility to services† | Major cities | 5286 | 65.41 | 7902 | 72.60 | 71574 | 68.82 |
| | Inner regional | 1684 | 20.84 | 1473 | 13.55 | 17646 | 16.97 |
| | Outer regional | 751 | 9.29 | 964 | 8.86 | 9623 | 9.25 |
| | Remote | 98 | 1.21 | 244 | 2.24 | 1454 | 1.40 |
| | Very remote | 250 | 3.09 | ‡ | ‡ | 3535 | 3.40 |
| | Unknown‡ | 12 | 0.15 | ‡ | ‡ | 177 | 0.17 |
| Bases of diagnosis | Non-pathological | 249 | 3.12 | 360 | 3.35 | 3413 | 3.32 |
| | Pathological | 7738 | 96.88 | 10398 | 96.65 | 99536 | 96.68 |
| Hospital catchment area | Other | 6742 | 83.43 | 9142 | 83.99 | 87384 | 84.02 |
| | Bentley and Armadale | 1339 | 16.57 | 1742 | 16.01 | 16625 | 15.98 |
| Number of comorbidities§ | None | 746 | 9.23 | 383 | 3.52 | 4854 | 4.67 |
| | 1–2 | 2502 | 30.96 | 1856 | 17.05 | 22136 | 21.28 |
| | 3–5 | 2773 | 34.32 | 3830 | 35.19 | 37740 | 36.29 |
| | 6+ | 2060 | 25.49 | 4815 | 44.24 | 39279 | 37.77 |
| Solid cancer | Cancer of the male genital organs | 1436 | 17.77 | 1830 | 16.81 | 19353 | 18.61 |
| | Cancer of the digestive organs | 1438 | 17.79 | 1753 | 16.11 | 18222 | 17.52 |
| | Melanoma | 1280 | 15.84 | 1891 | 17.37 | 16292 | 15.66 |
| | Breast cancer | 1087 | 13.45 | 1630 | 14.98 | 14262 | 13.71 |
| | Cancer of the respiratory organs | 679 | 8.40 | 875 | 8.04 | 8381 | 8.06 |
| | Cancer of the urinary tract | 322 | 3.98 | 511 | 4.69 | 4541 | 4.37 |
| | Cancer of the female genital organs | 348 | 4.31 | 438 | 4.02 | 4130 | 3.97 |
| | Mouth and pharynx cancer | 229 | 2.83 | 322 | 2.96 | 3049 | 2.93 |
| | Thyroid cancer | 148 | 1.83 | 238 | 2.19 | 2147 | 2.06 |
| | Brain cancer | 111 | 1.37 | 158 | 1.45 | 1523 | 1.46 |
| | Cancer of the soft tissues | 114 | 1.41 | 153 | 1.41 | 1459 | 1.40 |
| | Cancer of unspecified sites | 121 | 1.50 | 112 | 1.03 | 1318 | 1.27 |
| | Bone cancer | 9 | 0.11 | 12 | 0.11 | 157 | 0.15 |
| | All solid | 7322 | 90.61 | 9923 | 91.17 | 94834 | 91.18 |

Continued

**Table 1** Continued

| Characteristics | | Year of cancer diagnosis | | | | | |
|---|---|---|---|---|---|---|---|
| | | 2004 n=8081 | | 2014 n=10884 | | 2004–2014 (overall) n=104009 | |
| | | n | % | n | % | n | % |
| All haematopoietic cancer§ | Leukaemias and myelodysplasias | 327 | 4.05 | 330 | 3.03 | 3676 | 3.53 |
| | Other lymphomas | 203 | 2.51 | 292 | 2.68 | 2742 | 2.64 |
| | Other lymphoid cancers | 187 | 2.31 | 289 | 2.66 | 2280 | 2.19 |
| | Hodgkin's lymphoma | 42 | 0.52 | 50 | 0.46 | 477 | 0.46 |
| | All lymphoid and haematopoietic cancer | 759 | 9.39 | 961 | 8.83 | 9175 | 8.82 |

Indigenous status cannot be reported as per restrictions imposed by the human research ethics approval and can only be used for model adjustment purposes. Twelve months after the diagnosis is inclusive of the date of the diagnosis.
*Socioeconomic status measured by SEIFA-IRSD: Socio-Economic Indexes For Areas, Index of Relative Socio-Economic Disadvantage.
†Accessibility to services was measured using ARIA: Accessibility and Remoteness Index of Australia.
‡Where small cells are suppressed, random adjustments have been made to other cells within different levels of the same variable to prevent calculation of suppressed cell counts.
§Comorbidity: Number of comorbid conditions reported in hospitalisation data in the 5 years prior to the date of cancer diagnosis.
ED presentations, emergency department presentations.

### ED presentations and hospitalisation 12 months before and after cancer diagnosis and their trends over the study period

Similarly, to the pattern observed for CT scanning, as shown in table 2 when adjusted for year of diagnosis, the rate of hospitalisation was highest in the month of diagnosis at 910 per 1000 patients (95% CI 902 to 919) whereas for ED presentations the peak was in the month prior to diagnosis at 181 per 1000 patients (95% CI 178 to 184). While the rate dropped in the months after diagnosis for both ED presentations and hospitalisations, it remained consistently higher than the rate before diagnosis. (Further details of regression model output is presented in online supplemental appendix A).

As shown in figure 1B,C, no significant difference was observed in the rate of ED presentations or hospitalisation 12 months before cancer diagnosis in 2004 compared with 2014. However, significantly increased rates of both ED presentations and hospitalisations were observed in 2014 compared with those in 2004 for the month of diagnosis and thereafter. Similar patterns were observed for both cancer groups.

Further details in online supplemental appendix A show that patients with cancer diagnosis based on pathological methods were 34% less likely to have an ED presentation compared with those with cancer diagnosis based on non-pathological methods (RR 0.66 (95% CI 0.63 to 0.69)) whereas no significant difference was reported for hospitalisation rate. Increasing number of comorbidities was associated with substantially higher rates of both ED presentations and hospitalisations in which the RR between those with none versus those with at least six comorbidities was highest at 5.54 (95% CI 5.15 to 5.96) for ED presentations and 11.5 (95% CI 10.5 to 12.5) for hospitalisations. The rate of ED presentations also significantly increased with higher social disadvantage and with living in regional or remote areas. Small or no differences in hospitalisation and ED presentation rates were recorded in subgroups stratified by sex, age, cancer groups and hospital catchment region.

In figure 2B,C, over the entire study period the rate of both hospitalisation and ED presentations for those with ALH cancer were consistently higher than those with solid cancer both before and after diagnosis with little change in rate across the year of diagnosis (further details of regression model output is presented in online supplemental appendix C and D).

### DISCUSSION

Our study found that the rate of CT scanning started to increase from 2 months before diagnosis while ED presentations and hospitalisation started to increase from 1 month before diagnosis. These rates peaked in the month of diagnosis then rapidly reduced within the first month after diagnosis but remained high for the next 12 months. Our results were consistent with those reported in previous studies that showed an initial increase in the use of CT scanning or 'diagnostic investigation' (eg, X-ray, ultrasound, CT scan, MRI scan, angiography, endoscopies and biopsies) 2 months prior to cancer diagnosis.[19 21] Other studies have investigated use of CT scanning in specific cancer types with a UK study finding that rates of CT use started to increase 5 months before a diagnosis of bladder and kidney cancer,[17] and a Danish study finding an increased CT use associated with abdominal cancers diagnosed from 2014 to 2018.[18] In comparison our study evaluated changes in use of CT scanning associated with cancer diagnosis more globally and compared across a longer time period (2004–2014). The 10-year time frame means that our study period would have included more substantial changes in healthcare policy, cancer treatments and imaging technology that may have

**Table 2** Adjusted rate of CT scanning, ED presentations and hospitalisations in 12 months before and after cancer diagnosis

| Month before and after cancer diagnosis | CT scanning Rate/1000 patients (95% CI) | Hospitalisations Rate/1000 patients (95% CI) | ED presentations Rate/1000 patients (95% CI) |
|---|---|---|---|
| −12 | 14 (13 to 15) | 48 (46 to 50) | 26 (25 to 27) |
| −11 | 14 (13 to 15) | 51 (48 to 53) | 27 (25 to 28) |
| −10 | 14 (13 to 15) | 51 (49 to 53) | 27 (26 to 28) |
| −9 | 15 (14 to 15) | 52 (49 to 54) | 28 (26 to 29) |
| −8 | 15 (15 to 16) | 52 (50 to 55) | 28 (27 to 29) |
| −7 | 16 (15 to 17) | 56 (53 to 58) | 29 (28 to 30) |
| −6 | 18 (17 to 19) | 56 (53 to 58) | 30 (29 to 31) |
| −5 | 20 (19 to 21) | 58 (56 to 61) | 32 (31 to 33) |
| −4 | 24 (23 to 25) | 64 (62 to 67) | 37 (35 to 38) |
| −3 | 35 (33 to 36) | 73 (70 to 75) | 43 (42 to 45) |
| −2 | 69 (67 to 71) | 94 (91 to 96) | 59 (58 to 61) |
| −1 | 294 (290 to 298) | 371 (367 to 375) | 181 (178 to 184) |
| 1 | 477 (471 to 482) | 910 (902 to 919) | 125 (123 to 128) |
| 2 | 194 (190 to 197) | 624 (617 to 631) | 104 (102 to 106) |
| 3 | 150 (146 to 153) | 636 (629 to 644) | 94 (91 to 96) |
| 4 | 133 (130 to 136) | 587 (580 to 595) | 84 (82 to 87) |
| 5 | 120 (117 to 123) | 502 (496 to 509) | 75 (73 to 77) |
| 6 | 119 (116 to 122) | 431 (425 to 438) | 70 (68 to 72) |
| 7 | 119 (116 to 122) | 369 (363 to 375) | 63 (61 to 65) |
| 8 | 111 (108 to 114) | 312 (306 to 318) | 61 (59 to 63) |
| 9 | 100 (97 to 103) | 259 (254 to 264) | 57 (55 to 58) |
| 10 | 95 (92 to 98) | 236 (231 to 241) | 56 (54 to 58) |
| 11 | 93 (90 to 96) | 223 (218 to 228) | 53 (51 to 55) |
| 12 | 95 (92 to 97) | 214 (209 to 219) | 52 (50 to 54) |

Full model with adjusted covariates is presented in online supplemental appendix A.
ED, emergency department.

contributed to changes in CT.[32] While it was out of the scope of this study to evaluate the incremental impact of these changes, it is worth noting the extensive changes that have occurred over this time period that have cumulatively led to the changes in CT scanning observed.

In our study, although the rate of CT scanning was low at 4 months or more prior to the cancer diagnosis date, 30.4% of patients with cancer had the first CT scan performed at 4 months or more before the diagnosis date. Evidence shows that out-of-hospital referrals are more likely to be the source of early diagnosis using CTs,[33 34] whereas those referred as inpatients and emergency patients would be more likely to have more advanced cancers. Due to limited access to GPs and specialists or difficulty in funding out-of-hospital copayments, some Australian patients access primary care via EDs seeking diagnostic care.[35] Moreover, the likelihood of early CT use potentially could be driven by defensive diagnosis aimed to exclude later staging or advanced disease (whether it is the primary site or of metastatic nature).[36] This aligns with earlier evidence from the UK

of insufficient CT referrals for brain tumours, where existing guidelines do not adequately identify clinical symptoms useful for the basis of CT imaging.[37]

Our study is consistent with a study in New South Wales (NSW), Australia, that reported a similar pattern on the increase in rate of ED presentation and hospitalisation before diagnosis that dramatically reduced after diagnosis, although the NSW study was limited to admissions from ED only.[20] The observed increase in the use of CT scanning, ED presentations and hospitalisations in our study suggests a 'diagnostic time window' of 2 months whereas the increasing rate of general practice consultations have previously been found to start earlier, from 3 to 12 months before the diagnosis.[21 38 39] This time disparity between patients who ultimately are diagnosed with cancer initially presenting to their doctors and CT use could reflect issues with access to CT or indicate that CT is predominantly used for staging and guiding treatment direction post initial diagnosis. It also reflects that there may be a more complex diagnostic process with many steps prior to CT use. Since our study was limited to

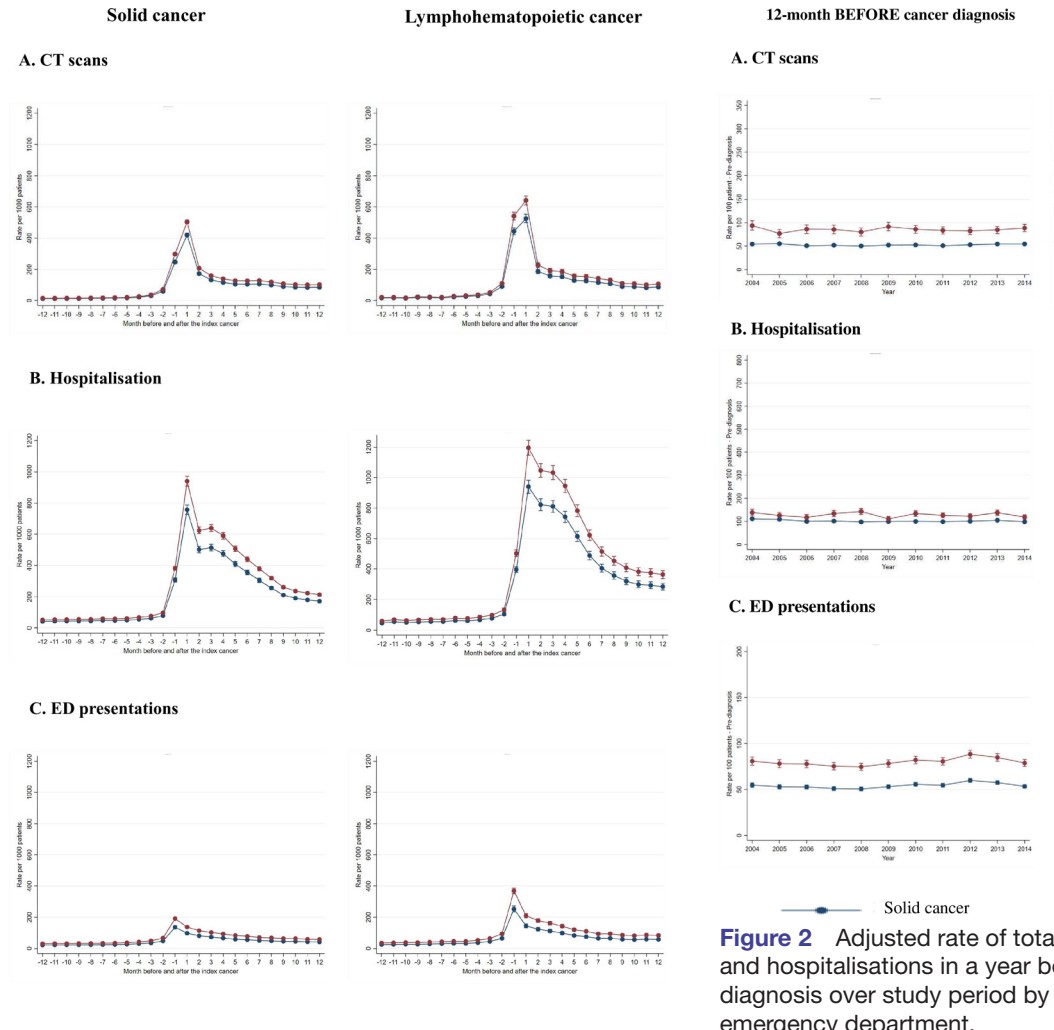

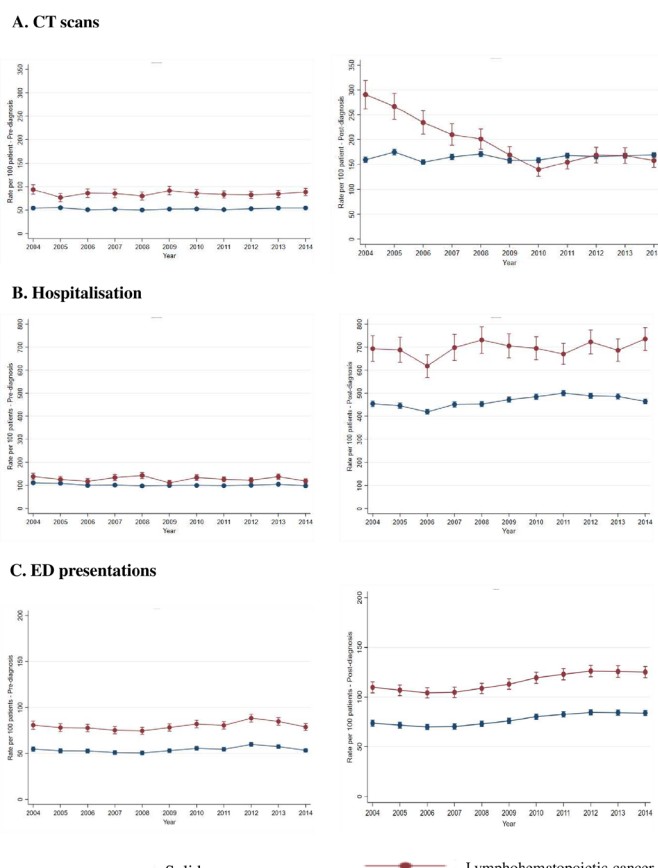

**Figure 2** Adjusted rate of total CT scans, ED presentations and hospitalisations in a year before and after cancer diagnosis over study period by cancer groups. ED, emergency department.

**Figure 1** Adjusted rate of CT scanning, ED presentations and hospitalisation in months before and after diagnosis by cancer groups in 2004 and 2014. ED, emergency department.

patients with a diagnosis of cancer, we cannot comment on the use of CT for those who were ultimately found not to have cancer but where CT used to rule this out.

Over the whole 10-year study period, patients with an ALH cancer were more likely to have CT scans than those with a solid cancer. This may be explained by the principal role of CT scans in staging patients with ALH cancer, particularly lymphomas[40–42] while solid cancers can be diagnosed by different imaging (eg, MRI, mammography, ultrasound), blood tests for biomarkers and biopsies that varies by cancer specific types. However, this hypothesis is not supported by our comparison of changes in CT scanning in 2004 compared with 2014, where we found that the rate of CT scanning reduced for patients with ALH cancer and increased for those with solid cancers. The reduction of CT scanning for patients with ALH cancer may be explained by the uptake of PET/CT scanning—a powerful tool for oncology which became available in WA in early 2002.[43] However, CT/PET scans are not captured by our data as it is limited to stand alone CT scanning,

therefore we cannot determine to what extent substitution between CT and PET/CT has taken place.

Also, we observed a higher rate of ED presentation in patients with cancer with higher disadvantage socioeconomic status and higher number of comorbidities that was in line with results from previous studies and reports.[44–46] In contrast to our study, an NSW report recorded a lower rate of ED presentations in those living in regional/remote areas as compared with those living in major cities.[46] The differences may be attributable to major factors that were not adjusted in our analysis such as location of hospitals, number of GP visits, stage at cancer diagnosis and other residual confounding.

### Strengths and limitations

This study included whole WA population incident cancer data linked to administrative records of health service use, which allows us to make grounded and representative conclusions for the target population based on higher statistical precision, avoidance of selection bias and negligible loss to follow-up. Yet the study has several limitations. First, lack of information on the indications for CT scans prevented us from determining if the CT scans were referred for cancer diagnosis or for other

health issues. Therefore, our findings have been reported in terms of use of CT within 12 months before and after a diagnosis of cancer and not in terms resulting from the cancer diagnosis per se. Second, CT scans undertaken at two secondary hospitals who contract out their radiology services are not included in the data, hence, the rate of CT use may be underestimated. However, since only a small proportion of the population reside in the two suburbs in question, and we adjusted for place of residence in the analysis (which was found to be non-significant), the missing data are likely to have minimal impact on the result. Although the data is several years old, to our knowledge this is one of the first studies fully examining the use of CT scans and healthcare services used around the diagnosis window at the population-based level. As such it provides valuable information for planning healthcare resources and to identify areas for further investigation of factors driving the growth of use of health services. The results will also provide valuable baseline data for future studies evaluating the impact of international campaigns (ie, Choosing Wisely) in improving appropriate use in diagnostic testing such as CT scanning.

## CONCLUSION

Our results showed an increase in demand for health services from 3 months before diagnosis of cancer. While the pattern of use before and after diagnosis remained unchanged for CT, ED and hospitalisations over the study period, differences in the magnitude of use were observed. From diagnosis onwards, the magnitude of the rate of ED presentations and hospitalisations was higher in 2014 compared with the baseline year of 2004 for both cancer types. In a comparison following diagnosis the rate of CT use was significantly higher in 2014 compared with the baseline year 2004 for solid cancer, however it was lower in ALH cancers. The persisting high use of CT during post cancer diagnosis and increased CT use observed in solid cancers may warrant a further investigation of factors driving this change and whether there is any potential over-testing in this group.

**Author affiliations**
[1]School of Population Health, Curtin University, Perth, Western Australia, Australia
[2]Sydney School of Health Sciences, University of Sydney, Sydney, New South Wales, Australia
[3]Nepean Blue Mountains Local Health District, New South Wales Health, Sydney, New South Wales, Australia
[4]Centre for Health Services Research, School of Population and Global Health, The University of Western Australia, Perth, Western Australia, Australia
[5]Biostatistics, University of Notre Dame, Fremantle, Western Australia, Australia
[6]Australian Women and Girls' Health Research (AWaGHR) Centre, Faculty of Medicine, University of Queensland, Brisbane, Queensland, Australia
[7]School of Physical Sciences, University of Adelaide, Adelaide, South Australia, Australia
[8]Obstetrics and Gynaecology Medical School, Faculty of Health and Medical Sciences, The University of Western Australia, Perth, Western Australia, Australia
[9]PathWest Laboratory Medicine, QE2 Medical Centre, Nedlands, Western Australia, Australia
[10]Fiona Stanley Hospital, Murdoch, Western Australia, Australia
[11]Division of Internal Medicine, Medical School, Faculty of Health and Medical Sciences, The University of Western Australia, Perth, Western Australia, Australia
[12]School of Medicine, College of Health and Medicine, University of Tasmania, Hobart, Tasmania, Australia
[13]Western Australian Cancer Registry, Clinical Excellence Division, Department of Health, East Perth, Western Australia, Australia

**Acknowledgements** We thank the National Health and Medical Research Council for supporting this work. We acknowledge the Western Australian Data Linkage Unit and custodians of the Hospital Morbidity Data Collection, the WA Cancer Registry, the Emergency Department Data Collection, the WA Death registry and the WA Picture Archiving and Communications System, as well as the individuals whose data enabled this study.

**Contributors** NTH, SK, MKB, JD, DM, DY, PO, RT and RM contributed to the study design and concept. RM, DY and NTH contributed to the acquisition of data. RM, SK, DY, CH, MKB, JD, DM, PO, CW, DM and NTH contributed to the analysis and interpretation of the data. NTH, CH, DY, SK and RM contributed to drafting of the manuscript. RM, JD, DM, PO, DY, CW, DM, RT and NTH contributed to critical revision of the manuscript for important intellectual content. RM and NTH contributed to statistical expertise. RM, MKB, JD, DM and PO secured funding for the study. The finished work has been read and approved by all authors.

**Funding** This study was funded with an Australian National Health and Medical Research Council grant, project grant APP1144573.

**Competing interests** The institutions of RM, NTH, PO, DY, CW, MKB and DM received grant funding from the National Medical Research Council of Australia for investigator-initiated research. The funding agreement ensured author independence in designing the study, interpreting the data, writing and publishing the report.

**Patient and public involvement** Patients and/or the public were involved in the design, or conduct, or reporting, or dissemination plans of this research. Refer to the Methods section for further details.

**Patient consent for publication** Not applicable.

**Ethics approval** Ethics approval was provided by the Curtin University Human Research Ethics Committee (HRE2017-0822), the Australian Institute of Health and Welfare Human Research Ethics Committee (EO2018/4/485) and the Western Australian Department of Health Human Research Ethics Committee (2011/97). The Human Research Ethics Committees approval was granted including a waiver of patient consent, since this was a whole-of-population study and no patients were contacted directly.

**Provenance and peer review** Not commissioned; externally peer reviewed.

**Data availability statement** Data may be obtained from a third party and are not publicly available. The data that support the findings of this study are available from the relevant data custodians of the study data sets. Restrictions by the data custodians mean that the data are not publicly available or able to be provided by the authors. Researchers wishing to access the data sets used in this study should refer to the Western Australian Data Linkage Unit and the Australian Institute of Health and Welfare.

**ORCID iDs**
Ninh Thi Ha http://orcid.org/0000-0002-2789-5604
Sviatlana Kamarova http://orcid.org/0000-0002-5271-8482
David Youens http://orcid.org/0000-0002-4296-4161

Chau Ho http://orcid.org/0000-0002-4285-9409
Max K Bulsara http://orcid.org/0000-0002-8033-6123
Rachael Moorin http://orcid.org/0000-0001-8742-7151

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
