## [Reviewer comments · BMJ Open]

ARTICLE DETAILS

TITLE (PROVISIONAL)	The use of computed tomography, ED presentation and hospitalisations 12-months before and after a diagnosis of cancer in Western Australia: A population-based retrospective cohort study
AUTHORS	Ha, Ninh; Kamarova, Sviatlana; Youens, David; Ho, Chau; Bulsara, Max; Doust, Jenny; Mcrobbie, Donald; O'Leary, Peter; Wright, Cameron; Trevithick, Richard; Moorin, Rachael

VERSION 1 – REVIEW

REVIEWER	Tim Sadler Cambridge University
REVIEW RETURNED	22-Apr-2023

GENERAL COMMENTS	Thank you producing this study. It is well written, based upon a large data set of patients over a ten year period and is interesting to read. Page 11 Lines 21-23: "When comparing the magnitude of CT rates in 2004 with those in 2014, higher rates were observed from 1 month prior to diagnosis onwards in solid cancers, whereas CT scanning rates reduced in ALH cancer." - Can you please clarify this. This paragraph refers to Figure 1. But Figure 1 panel A shows higher rates for CTs in ALH cancers in 2014 than for 2004? Regarding the discussion points on CT scanning reducing for patients with ALH cancer between 2004 and 2014 - this is more likely accounted for by PET-CT being used for staging of ALH (especially lymphoma) instead of standard CT, the use of which has likely increased since 2004 in your study population. Hence the decline in CTs. (I note that PET-CTs were excluded in the methodology). The data set is several years old now. The number of CT studies performed typically increases by 10% each year. Have you considered discussing any potential limitations of this and are the results still relevant? Line 14-15 of conclusion: "The persisting high use of CT during post cancer diagnosis and increased CT use observed in solid cancers may warrant a further investigation of factors driving this change." - This is due to
---

REVIEWER	Bonnie E. Gould Rothberg Yale Comprehensive Cancer Center
REVIEW RETURNED	22-Apr-2023

GENERAL COMMENTS	I wish to thank the authors for allowin
---

	Major comments 1. The study covers 11 years of data. One of the potential confounders is the introduction of secular trends in the use of CT scans during the study period. For example, screening for lung cancer Using CT scans began towards the end of the study period. It would be important for the investigators to adjust for time/year or data collection. Although the language of he results suggests that this might have been done, it is not clearly, up front stated in the methods. The authors allude to this bias in their Discussion as well but do not address how they account for it in their methods. 2. Given that “region” is one of the covariates and that data for each timepoint within each region have a structure of correlation, have the authors considered using a Longitudinal Data Analysis approach with random effects for intercept (region) and slope (time) to increase the robustness of the model? A negative binomial distribution can be fit with a unstructured correlation as the most conservative measure. 3. In their Discussion, it would be interesting for the authors to explore reasons why there is an increase in CT scanning as far out as 90 months pre-diagnosis. Is there the possibility for latent/occult symptoms that warrant work-up but yet yield negative results on imaging (otherwise the scan would have led to the diagnosis and triggered t=0)? Is there precedent in the literature for this observation? 4. The authors show that the rate for CT scanning is higher for lymphohematopoeitic cancers compared with solid tumors. To me, this is counter-intuitive as leukemias usually manifest as new anemias detected incidentally on routine health maintenance or as thrombocytopenias in the setting of easy bruising. It would stand that only lymphomas would have symptomatology that would be consistent with an indication for CT scanning. Given the large sample size, there should be enough power to conduct a stratified analysis according to cancer type with major groupings for lymphoma, GI cancers, thoracic cancers, genitourinary, gynecologic, melanoma, sarcoma, etc. A 10-12-level variable might be interesting as a secondary analysis. Minor comments: 1. Please define AIHW on its first instance of use (Page 6, Line 9)
--	---

VERSION 1 – AUTHOR RESPONSE

Reviewer: 1

Dr. Tim Sadler , Cambridge University

Comments to the Author:

Thank you producing this study. It is well written, based upon a large data set of patients over a ten year period and is interesting to read.

Author response:

Thank you very much for your positive feedback.

Page 11 Lines 21-23: "When comparing the magnitude of CT rates in 2004 with those in 2014, higher rates were observed from 1 month prior to diagnosis onwards in solid cancers, whereas CT scanning rates reduced in ALH cancer." - Can you please clarify this. This paragraph refers to Figure 1. But Figure 1 panel A shows higher rates for CTs in ALH cancers in 2014 than for 2004?

Author response:

Thank you for pointing this out. We agree this sentence does not accurately reflect the results in figure 1 panel A. We have revised the sentence to be more appropriate as follows.

Page 16, line 5: "For both years a peak in CT use occurred in the month of diagnosis and the peak was more pronounced for ALH cancers than solid cancer"

Regarding the discussion points on CT scanning reducing for patients with ALH cancer between 2004 and 2014 - this is more likely accounted for by PET-CT being used for staging of ALH (especially lymphoma) instead of standard CT, the use of which has likely increased since 2004 in your study population. Hence the decline in CTs. (I note that PET-CTs were excluded in the methodology).

Author response:

Thank you for your comments. We agree. The introduction of PET/CT scanning in Western Australia in 2002 could potentially explain the reduction in the CT scanning observed in our study period between 2004 and 2014. We have incorporated this into the discussion as follows.

Page 19, line 24: "The reduction of CT scanning for patients with ALH cancer may be explained by the uptake of PET/CT scanning – a powerful tool for oncology which became available in Western Australia in early 2002 (Lau et al., 2005). However, CT/PET scans are not captured by our data as it is limited to stand alone CT scanning, therefore we cannot determine to what extent substitution between CT and PET/CT has taken place."

The data set is several years old now. The number of CT studies performed typically increases by 10% each year. Have you considered discussing any potential limitations of this and are the results still relevant?

Author response:

Although the data is several years old, to our knowledge this is one of the first studies fully examining the use of CT scans and health care services used around the diagnosis window at the population-based level. As such it provides valuable information for planning health care resources and to determine areas to be further investigated regarding the factors driving the growth of health services used. In addition, increasing use of CT scanning has sparked concerns of over-testing. Over the past decade, there have been international campaigns such as Choosing Wisely implemented to improve appropriate use of diagnostic tests including CT scanning. The campaigns were initiated in United States and followed by Canada and then launched in Australia in 2015. The campaigns aimed to reduce unnecessary treatment and diagnostic testing, including specific recommendations for the use of CT scanning. Although our data do not cover the period after the implementation of Choosing Wisely in Australia, the results will provide valuable baseline data for future studies evaluating impact of such programs on the use of CT scanning. We have revised the discussion section to include the limitations and relevant of the findings in the future as follows.

Page 20, line 24: "Although the data is several years old, to our knowledge this is one of the first studies fully examining the use of CT scans and health care services used around the diagnosis window at the population-based level. As such it provides valuable information for planning health care resources and to identify areas for further investigation of factors driving the growth of use of health services. The results will also provide valuable baseline data for future studies evaluating the impact of international campaigns (i.e., Choosing Wisely) in improving appropriate use in diagnostic testing such as CT scanning."

Line 14-15 of conclusion: "The persisting high use of CT during post cancer diagnosis and increased CT use observed in solid cancers may warrant a further investigation of factors driving this change." - This is due to

Author response:

We have revised the sentence as follows, hope that makes it clearer.

Page 21, line 16: "The persisting high use of CT during post cancer diagnosis and increased CT use observed in solid cancers may warrant a further investigation of factors driving this change and whether there is any potential over-testing in this group."

Reviewer: 2

Dr. Bonnie E. Gould Rothberg, Yale Comprehensive Cancer Center
Comments to the Author:

Major comments

1. The study covers 11 years of data. One of the potential confounders is the introduction of secular trends in the use of CT scans during the study period. For example, screening for lung cancer Using CT scans began towards the end of the study period. It would be important for the investigators to adjust for time/year or data collection. Although the language of the results suggests that this might have been done, it is not clearly, up front stated in the methods. The authors allude to this bias in their Discussion as well but do not address how they account for it in their methods.

Author response:

Thank you for your comment. We have adjusted for time using a year variable in all regression models. We have revised the method section as follows to make this clear.

Page 8, line 14: "Numbers of CT scans, hospitalisations and ED presentations were captured monthly and annually for all cancer diagnosis in the study period from 2004 to 2014."

Page 9, line 17 and 22: ".....adjusted for all observed socio-demographic and clinical characteristics and year of diagnosis. Multivariable negative binomial regression was also used to examine trends in total CT scanning use, ED presentations and hospitalisations in the year prior to and after diagnosis over the study period (2004 to 2014) adjusted for all observed characteristics and year of diagnosis."

2. Given that "region" is one of the covariates and that data for each timepoint within each region have a structure of correlation, have the authors considered using a Longitudinal Data Analysis approach with random effects for intercept (region) and slope (time) to increase the robustness of the model? A negative binomial distribution can be fit with an unstructured correlation as the most conservative measure.

Author response:

Thank you for your comment. We assume you are referring to the variable "accessibility to services". We agree the accessibility may have correlation with time in year variable. Random effects are used for controlling unobserved heterogeneity which are not correlated with observed explanatory variables (Wooldridge, 2010). However, since they are potentially correlated, treating "accessibility" as a random effect in the model is not appropriate.

3. In their Discussion, it would be interesting for the authors to explore reasons why there is an increase in CT scanning as far out as 90 months pre-diagnosis. Is there the possibility for latent/occult symptoms that warrant work-up but yet yield negative results on imaging (otherwise the scan would have led to the diagnosis and triggered t=0)? Is there precedent in the literature for this observation?

Author response:

Thank you for your comments. To clarify our study evaluated CT use within 12 months prior and post cancer diagnosis date. In our study, the rate of CT scanning was low for the period of above 3 months ~ 90 days (not months) prior to the cancer diagnosis date, and a total of 30.4% of cancer patients had the first CT scan performed 90 days or more before the diagnosis date. Early CTs were more likely to be referred out-of-hospital and less likely to be in-patient referral-based, compared to emergency referrals. Indeed, evidence shows that out-of-hospital referrals are more likely to be the source of early diagnosis using CTs (Guldbrandt, 2015; Møller et al., 2019), whereas those referred as in-patients and emergency patients would be more likely to have more advanced cancers. Due to limited access to general practitioners and specialists or difficulty in funding out-of-hospital co-payments, some Australian patients access primary care via EDs seeking diagnostic care (Lowthian et al., 2011). Moreover, the likelihood of early CT use potentially could be driven by defensive diagnosis aimed to exclude later staging or advanced disease (whether it is the primary site or of metastatic nature) (Mohr et al., 2009). This aligns with earlier evidence from the UK of insufficient CT referrals for brain

tumours, where existing guidelines do not adequately identify clinical symptoms useful for the basis of CT imaging (Zienius et al., 2019).

We have revised the discussion section to include this point as follows.

Page 18, line 17: "In our study, although the rate of CT scanning was low at 4 months or more prior to the cancer diagnosis date, 30.4% of cancer patients had the first CT scan performed at 4 months or more before the diagnosis date. Evidence shows that out-of-hospital referrals are more likely to be the source of early diagnosis using CTs (33, 34), whereas those referred as in-patients and emergency patients would be more likely to have more advanced cancers. Due to limited access to general practitioners and specialists or difficulty in funding out-of-hospital co-payments, some Australian patients access primary care via EDs seeking diagnostic care (35). Moreover, the likelihood of early CT use potentially could be driven by defensive diagnosis aimed to exclude later staging or advanced disease (whether it is the primary site or of metastatic nature) (36). This aligns with earlier evidence from the UK of insufficient CT referrals for brain tumours, where existing guidelines do not adequately identify clinical symptoms useful for the basis of CT imaging (37)."

4. The authors show that the rate for CT scanning is higher for lymphohematopoietic cancers compared with solid tumors. To me, this is counter-intuitive as leukemias usually manifest as new anemias detected incidentally on routine health maintenance or as thrombocytopenias in the setting of easy bruising. It would stand that only lymphomas would have symptomatology that would be consistent with an indication for CT scanning. Given the large sample size, there should be enough power to conduct a stratified analysis according to cancer type with major groupings for lymphoma, GI cancers, thoracic cancers, genitourinary, gynecologic, melanoma, sarcoma, etc. A 10-12-level variable might be interesting as a secondary analysis.

Author response:

Thank you for your suggestion. We have included the results showing the trends of CT use across cancer types in appendix D for additional information. We have also revised the statistical analysis, and results to include the change as follows.

Page 9, line 17: "Similarly, monthly rates of CT scanning, ED presentations and hospitalisation 12 months prior to and post cancer diagnosis were conducted for sub-cancer types and the results are presented in Appendix D."

Page 14, line 10: "Similar patterns were observed across sub-cancer types as shown in Appendix D."

Minor comments:

1. Please define AIHW on its first instance of use (Page 6, Line 9)

Author response:

We have revised the abbreviation as suggested.

VERSION 2 – REVIEW

REVIEWER	Tim Sadler Cambridge University
REVIEW RETURNED	08-Jul-2023
GENERAL COMMENTS	Thank you for responding and appropriately addressing the comments of my review. I have also reviewed the responses and changes made regarding comments of Reviewer 2 and have nothing additional to add.